# Intrinsic Flexibility of the EMT Zeolite Framework under Pressure

**DOI:** 10.3390/molecules24030641

**Published:** 2019-02-12

**Authors:** Antony Nearchou, Mero-Lee U. Cornelius, Jonathan M. Skelton, Zöe L. Jones, Andrew B. Cairns, Ines E. Collings, Paul R. Raithby, Stephen A. Wells, Asel Sartbaeva

**Affiliations:** 1Department of Chemistry, University of Bath, Claverton Down, Bath BA2 7AY, UK; a.nearchou@bath.ac.uk (A.N.); jonathan.skelton@manchester.ac.uk (J.M.S.); zlj20@bath.ac.uk (Z.L.J.); chsprr@bath.ac.uk (P.R.R.); 2Department of Chemistry, University of the Western Cape, Bellville, Cape Town 7535, South Africa; meroleecornelius@gmail.com; 3School of Chemistry, University of Manchester, Oxford Road, Manchester M13 9PL, UK; 4Department of Materials, Imperial College London, Kensington, London SW7 2AZ, UK; a.cairns@imperial.ac.uk; 5European Synchrotron Radiation Facility, 71 avenue des Martyrs, 38000 Grenoble, France; ines.collings@esrf.fr; 6Department of Chemical Engineering, University of Bath, Claverton Down, Bath BA2 7AY, UK; saw42@bath.ac.uk

**Keywords:** zeolite, framework materials, EMC-2, crystallization, high pressure, X-ray diffraction, flexibility window, compressibility, lattice dynamics

## Abstract

The roles of organic additives in the assembly and crystallisation of zeolites are still not fully understood. This is important when attempting to prepare novel frameworks to produce new zeolites. We consider 18-crown-6 ether (18C6) as an additive, which has previously been shown to differentiate between the zeolite EMC-2 (EMT) and faujasite (FAU) frameworks. However, it is unclear whether this distinction is dictated by influences on the metastable free-energy landscape or geometric templating. Using high-pressure synchrotron X-ray diffraction, we have observed that the presence of 18C6 does not impact the EMT framework flexibility—agreeing with our previous geometric simulations and suggesting that 18C6 does not behave as a geometric template. This was further studied by computational modelling using solid-state density-functional theory and lattice dynamics calculations. It is shown that the lattice energy of FAU is lower than EMT, but is strongly impacted by the presence of solvent/guest molecules in the framework. Furthermore, the EMT topology possesses a greater vibrational entropy and is stabilised by free energy at a finite temperature. Overall, these findings demonstrate that the role of the 18C6 additive is to influence the free energy of crystallisation to assemble the EMT framework as opposed to FAU.

## 1. Introduction

Zeolites are crystalline inorganic polymers, characterised by their open framework structure and microporosity. Typically, zeolites are aluminosilicates, however the term is broadly used to describe materials of varying chemistry with zeolitic topologies. They are commonly used as molecular sieves, and due to the presence of strongly acidic localised active sites they can also be used for catalysis, ion exchange, and gas adsorption and separation, among other applications [1,2,3,4]. The basic structure comprises of TO_4_ tetrahedra (T = Si or Al) interconnected via the apical oxygen atoms. The arrangement of the primary tetrahedra into regular geometric subunits known as secondary building units (SBUs) defines the distinctive channel and cage structures of each zeolite topology [5,6]. The TO_4_ units themselves can be considered as almost rigid polyhedra, whereas the inherent flexibility in the T–O–T linkages allows the framework to respond to thermodynamic stimuli such as temperature and pressure [1,4,7,8].

The crystallisation of zeolites often involves the use of structure-directing agents (SDAs) that guide the assembly of the framework structural units through electrostatic and van der Waals (dispersion) interactions [9]. Typical SDAs are either metal cations [10] or organic additives [11,12], which during synthesis involve clathration of the anionic TO_4_ species in solution [13,14]. Although there is a broad understanding of the influence of SDAs on the synthetic outcome, little is known on the more subtle intricacies of how organic additives directly propagate specific zeolite frameworks. This is of particular interest in cases where a single organic additive is capable of assembling a number of topologically distinct zeolites [15]. Current approaches to understanding the structure-directing roles of organic additives typically assume pure geometric templating, where there is a level of symmetric and molecular recognition between the additive and an SBU or cage in the framework [12,16,17,18]. However, it is known that organic additives do not behave solely as geometric templates, but can influence crystallisation by a multitude of routes including space-filling [12] and inhibition of competing phases [19,20].

To date, there has been little investigation into the intrinsic flexibility of zeolites, which is an integral factor in both determining key properties for applications and in the assembly of zeolite structures during synthesis [8]. Current research has shown that all reported zeolites possess a “flexibility window”, defined as the range of volumes over which the framework can contract and expand through variation of the T–O–T angles while the TO_4_ tetrahedra remain rigid. It has thus been suggested that the existence of such a window is a necessary criterion for identifying hypothetical frameworks that are synthetically feasible [4,21]. Experimentally, framework flexibility can be probed by applying external pressure and studying the response of the structure. The mechanical behaviour of zeolites can be strongly influenced by the extra-framework content, such as occluded structure-directing cations [7,22,23,24] and organic additives [25], and connections between framework flexibility and guest structure-directing species can in principle provide valuable insights into the nature of the TO_4_ clathration process during crystallisation.

Previous geometric simulations using the GASP software [26,27] have suggested that the presence of 18-crown-6 ether (18C6) in the cavities of the EMT framework does not impede its intrinsic geometric flexibility [25], putting into question its role as a template. In this framework, the 18C6 guest occupies the *t-wof* supercage, as determined by Baerlocher et al. [28] and shown in Figure 1. In the Baerlocher crystal structure the 18C6 species express disorder, whereby there is a superposition of two molecules that are associated by a mirror plane and orientated in opposing directions along the *c* axis. However, based on the occupancy of the crystallographic sites there is only a single 18C6 molecule per *t-wof* cavity. To illustrate this, Figure 1 shows the two *t-wof* cavities in the unit cell occupied by a single 18C6 species which is in one of the two available orientations. This reduced symmetry structure has been used previously as the input for geometric simulations [25].

Current experimental research has highlighted the importance of 18C6 as an organic additive in the assembly of the EMT framework during synthesis, beginning with the formation of the [(18C6)Na^+^] macrocation in the hydrogel [28,29,30]. This macrocation sits in depressions on the surface of the growing crystal and electrostatically assembles the new SBU faujasite layer into the hexagonal EMT arrangement [31,32,33]. In the presence of excess Na^+^ cations, the growing framework is less siliceous, and faujasite (FAU)-type zeolites preferentially crystallise [31,34]. This strongly suggests a competition between the FAU and EMT phases during crystallisation. Simulation results further suggest that the macrocation influences the free-energy landscape during the crystallisation process rather than acting as a geometric template [25].

We carried out a high-pressure powder X-ray diffraction study on the EMT framework zeolite EMC-2, both with and without 18C6 occluded within the framework cavities. From this data, we report the first experimental measurements of the intrinsic flexibility of the EMT framework, which we find to be in good agreement with prior predictions from geometric simulations. We also performed first-principles calculations, including the effects of lattice dynamics on the free energy, to compare the stabilities of the competing FAU and EMT phases. Together, these results provide new insight into the role of 18C6 in defining the selectivity of the growth, and pose fundamental questions to be explored in future studies.

## 2. Results and Discussion

### 2.1. High-Pressure Powder X-ray Diffraction

Figure 2 shows the variation in lattice parameters of the filled and empty (i.e., with and without 18C6) zeolite EMC-2 as a function of pressure. The flexibility window of the EMT framework obtained from geometric simulations, which suggest the window to be identical for both the empty and 18C6 filled zeolite EMC-2, is overlaid for comparison [25]. This comparison indicates that our measurements were performed well within the window boundaries and, consequently, no distortions of the TO_4_ tetrahedra are anticipated over the range of pressures studied. For both the empty and filled samples, the *a* and *c* lattice parameters decrease with pressure, indicative of isotropic compression. Contraction of the unit cell proceeds in the same direction within the flexibility window regardless of whether the 18C6 molecule is present in the framework cavities. The onset of pressure-induced amorphisation, indicated by a broadening of the Bragg peaks in the diffraction patterns (see Appendix A), was observed for both samples before the edge of the flexibility window.

The unit cell parameters *a* and *c* are shown as a function of pressure in Figure 3a,b. It is apparent that the 18C6-containing zeolite can tolerate higher pressure before the onset of pressure-induced amorphisation, indicating that the 18C6 guest “braces” the framework and maintains structural integrity, delaying the onset of amorphisation. The effect of extra-framework channel content resisting framework collapse is anticipated and has been seen in sodalite containing bromide anions [24] as well as in silicalites [35].

From Figure 3a, it can be seen that the contraction of the *a* axis with increasing pressure is consistent between the empty and filled zeolites. In both cases, the contraction is linear with pressure up to 3 GPa, above which the *a* parameter of the empty EMC-2 decreases suddenly; this is concomitant with a sharp reduction in crystallinity and quality of the diffraction pattern (as shown in Appendix A). Such a decline in crystallinity is indicative of the onset of pressure-induced amorphisation and consequently structural perturbations at this pressure are expected. For the filled zeolite EMC-2, which resists framework collapse, the size of the contraction in the *a* axis decreases under increasing pressure, demonstrating reduced compressibility.

The contraction of the *c* axis (Figure 3b) is similarly comparable between the empty and filled zeolites. Although the *c* parameter in the filled zeolite is marginally more resistant to contraction over the 1–3 GPa region of pressures, the difference is likely not due to the presence of the 18C6, but rather to the different pressure-transmitting media employed in the two sets of measurements. The reason is that silicone oil, used for the filled zeolite, is known to lose its hydrostatic behavior at approximately 0.9 GPa [36]. This is emphasised in Appendix A, where repeated measurements indicate that this anisotropy is present when silicone oil is used as the pressure transmitting medium. We note that this deviation does not influence the overall cell volume. Despite these differences, the *c* axis of the filled zeolite shows decreasing compressibility under increasing pressure, again indicating pressure-induced stiffening, as with the *a* axis.

Despite subtle differences in the contraction of the *c* axis with the occlusion of the 18C6 molecule, the overall effect of the guest on the reduction in the cell dimensions under pressure is negligible. This can be clearly seen in the variation of the cell volume as a function of pressure (Figure 4), which demonstrates that the net compression of the framework with pressure is very similar for both the filled and empty zeolites. The figure also shows the bulk moduli of the empty and filled zeolite EMC-2 calculated using the PASCal web application from Cliffe and Goodwin [37]. These calculations were performed by fitting to the 2nd-order Birch–Murnaghan equation (see Appendix A). There is no significant difference in the bulk moduli between the empty and filled structures, further evidencing that occupation of the framework cages by the 18C6 molecule does not materially impact the compressibility.

Within the scope of open-framework silicates, the bulk moduli for both analogues of zeolite EMC-2 fall within the anticipated 15–70 GPa range [8]. Furthermore, the values are comparable to those measured for the high-silica zeolite Y (FAU) in a non-penetrating pressure-transmitting medium [38]. As both the FAU and EMT topologies share the same SBUs, this implies that the nature of zeolite compressibility is dictated by framework structural features.

Moreover, it is apparent that the intrinsic compressibility is not impeded by certain extra-framework content. The key implication of the similarity in mechanical behaviour is that it confirms previous predictions that the 18C6 molecule should not limit the geometric flexibility of the EMT framework [25]; although the 18C6 molecule is sterically bulky, the inherent flexibility of the molecule permits it to adjust to geometric deformations of the *t-wof* cavity it occupies. These data hint at the role of 18C6 in the assembly of the EMT framework during crystallisation. The lack of steric effects on the compressibility of the framework suggest that the 18C6 molecule likely does not behave as a geometric template. This is congruent with previous suggestions that the 18C6 organic additive instead subtly influences the free-energy landscape during the crystallisation process, promoting the assembly of the EMT framework [25]. This agrees with the current understanding of the synthesis mechanism [28,29,30,31,32,33,34].

### 2.2. Computational Modelling

To further investigate the inferences drawn from our experimental measurements, we used solid-state density-functional theory (DFT) calculations to compare the energies of the idealised FAU and EMT framework structures [39].

With the dispersion-corrected PBEsol + D3 functional, the optimised lattice parameters are an excellent match for the experimental measurements, with differences in the lattice parameters and cell volume of <1% (see Appendix A). Energy/volume curves fitted to the Birch–Murnaghan equation over volume compressions and expansions of ±5% about the calculated equilibrium yield bulk moduli of 41.3 and 39.2 GPa for FAU and EMT, respectively (see Appendix A), the latter of which is in near-quantitative agreement with our experimental measurements. A comparable bulk modulus of 40 was obtained for FAU by the alternative method of calculating the full elastic-constant matrix (see Appendix A).

Having confirmed that our calculations reproduce the measured structural properties of the two frameworks to a high degree of accuracy, we proceeded to compare the energetic stabilities of the EMT and FAU frameworks. To do so, we calculated the reaction energy ΔE for the hypothetical solid-state phase transformation:EMT_(s)_ → 2 FAU_(s)_(1)The factor of 2 arises from the FAU primitive cell containing half the number of SiO_2_ formula units (48) compared to EMT (96). We calculated a ΔE of –2.59 kJ mol^−1^ per EMT formula unit (96 T atoms; 2.69 × 10^−2^ kJ mol^−1^ per SiO_2_ formula unit) which indicates that, based on the lattice energy, FAU is the more stable of the two zeolite frameworks.

Calculated energy differences can however be highly sensitive to the choice of DFT functional. To test this, we performed additional calculations using “bare” PBEsol without the dispersion correction applied. The optimised lattice parameters from these calculations (see Appendix A) show larger deviations from the experimental measurements, but the size of the difference is consistent for both frameworks. These calculations predict a reaction energy of −5.69 kJ mol^−1^ (5.93 × 10^−2^ kJ mol^−1^ per SiO_2_ F.U.) which is approximately two times larger than with PBEsol + D3 but nonetheless indicates the same qualitative stability ordering.

Since both zeolites are formed in the presence of water, we also tested the effect of using the implicit-solvent VASPsol model [40,41] to mimic the dielectric environment of water in the cages (ε = 80.1). This made a minimal quantitative difference to the optimised lattice parameters (see Appendix A) but selectively stabilises EMT, resulting in calculated reaction energies of 2.28 and −1.28 kJ mol^−1^ (2.37 × 10^−2^ and −1.34 × 10^−2^ kJ mol^−1^ per F.U.) for PBEsol + D3 and PBEsol, respectively. This suggests that the growth conditions and/or the presence of molecules in the cages can significantly influence the energy differences between polymorphs and may render EMT the thermodynamic product.

As noted in previous studies [42,43], subtle differences in lattice dynamics and vibrational contributions to the free energy can also adjust the stability ordering of competing polymorphs at finite temperature. To investigate this, we performed lattice dynamics calculations with PBEsol + D3 on the FAU and EMT frameworks, from which we evaluated the temperature-dependent constant-volume (Helmholtz) free energy A=U−TS according to: [44](2)$$A\left(T\right)=U_{\mathrm{latt}}+\frac{1}{N}\left\{\frac{1}{2}\underset{qv}{\sum }\hbar \omega \left(qv\right)+k_{B}T\underset{qv}{\sum }\ln\left[1-\exp\left(\frac{-\hbar \omega \left(qv\right)}{k_{B}T}\right)\right]\right\}$$where $k_{B}$ is the Boltzmann constant, *U*_latt_ is the lattice internal energy, ω(qv) are the phonon frequencies at wavevector q and band index v, and N is the number of wavevectors included in the summation over reciprocal space. Of the three terms in Equation (2), the lattice energy is the total energy obtained from the DFT calculations, the second corresponds to a sum of the zero-point energies of the phonon modes, and the third term captures the temperature dependence of the internal energy and entropy due to population of phonon modes at finite temperature.

Figure 5 compares the calculated free energy ΔA of the EMT → FAU transformation in Equation (1) as a function of temperature. At 0 K, differences in the vibrational zero-point energy lead to a small increase in the reaction energy to −2.50 kJ mol^−1^ (−2.60 × 10^−2^ kJ mol^−1^ per F.U.; i.e., destabilising EMT with respect to FAU), but the effect is negligible. However, at higher temperatures differences in the vibrational free energy outweigh the difference in lattice energy and render EMT the more stable polymorph, with a free-energy difference of +2.39 kJ mol^−1^ (2.49 × 10^−2^ kJ mol^−1^ per F.U.) at 300 K. A decomposition of ΔA into ΔU and −TΔS terms shows clearly that this stabilisation arises from the higher vibrational entropy of the EMT framework compared to FAU, a conclusion in line with similar studies on other materials [42,43].

The phonon density of states (DoS) curves for the EMT and FAU frameworks (see Appendix A) show very similar features, with only subtle differences in the fine structure of the major peaks. Given that the structural differences between zeolites arises from the arrangement of a common structural unit (i.e., SiO_4_ tetrahedra) into larger SBUs, differences in the spectra of vibrational frequencies are expected to be most pronounced for low-frequency collective vibrations involving multiple tetrahedra, so this is not unexpected. However, for the same reasons, we might anticipate these differences to play an important role in defining the zeolite free-energy landscape.

The energetics calculations including lattice dynamics suggest that the EMT framework may be more stable than FAU at finite temperature. If taken at face value, this poses an interesting question: it may be that FAU-type zeolites are a kinetic product, and the role of the 18C6 organic additive in the reaction is to block its formation and allow zeolite EMC-2 to form. This is consistent with the observation that zeolite EMC-2 takes significantly longer than FAU-type zeolites to form [15,45].

We leave the task of validating (or otherwise) this suggestion to future experimental work, although we note that differences in the simulated IR spectra between 300–500 cm^−1^ (see Appendix A) suggest that it may in principle be possible to use vibrational spectroscopy to monitor the formation and growth of EMT- and FAU-type crystallites during synthesis. Whether this is ultimately possible will depend on a number of factors, chief among which is the presence of spectral signals from the solvent and/or 18C6 molecules.

Finally, we also note that the differences in vibrational free energy are likely to be additive to the solvent effects discussed above, i.e., the effect of the solvent/guest molecules and free-energy differences will combine to more strongly favour crystallisation of EMT over FAU.

## 3. Materials and Methods

### 3.1. Sample Preparation

Zeolite EMC-2 (EMT) was prepared following the procedure and batch composition reported previously [34]. Both the filled and empty structures (i.e., with and without 18C6) were prepared using this procedure. The molar batch composition of the precursor hydrogel was: 1.00 Al_2_O_3_/1.96 Na_2_O/9.68 SiO_2_/0.47 (18C6)/87.0 H_2_O. All materials were purchased from Sigma-Aldrich (Gillingham, Dorset, UK), viz. sodium hydroxide (NaOH), 18-crown-6 ether (C_12_H_24_O_6_; 18C6), sodium aluminate (NaAlO_2_), colloidal silica (LUDOX^®^ HS-40, 40 wt% SiO_2_ in water) and distilled water.

Sodium hydroxide and 18C6 were dissolved in the distilled water. The sodium aluminate was then added, and the solution stirred until homogenous. The colloidal silica was then added slowly to prevent rapid gelation. The resulting hydrogel was aged for 24 h under stirring before being transferred to a Teflon cup in a stainless-steel autoclave. The sealed autoclave was then kept in a 110 °C oven for 12 days to ensure complete crystallisation. The autoclave was then removed, opened, and the product separated from the mother liquor via filtration and washed until the filtrate was of neutral pH. The white residue was dried and ground for further preparation steps.

To produce the empty zeolite EMC-2, the powder was calcined in air. The sample was heated to 100 °C, 200 °C and 300 °C for 1 h, and finally to 450 °C for 6 h, at a ramp rate of 1 K min^−1^. After calcination, the sample was cooled at a rate of 1 K min^−1^, held at 200 °C for 1 h, and then cooled to ambient temperature.

Both the empty and filled zeolite EMC-2 samples were dehydrated under vacuum. The samples were heated from ambient temperature at a ramp rate of 1 K min^−1^, with static temperature segments at 100 °C for 1 h and 200 °C for 6 h. After dehydration, the sample was cooled at a rate of 1 K min^−1^ and held static at 100 °C for 1 h before cooling to ambient temperature.

### 3.2. High-Pressure Powder X-ray Diffraction

Empty and filled zeolite samples were analysed using high-pressure powder X-ray diffraction on the ID15B and ID27 beamlines, respectively, at the European Synchrotron Radiation Facility (ESRF) in Grenoble, France. The samples were loaded into diamond-anvil cells (DACs) using Daphne 7373 oil and silicone oil, respectively, as a non-penetrating pressure-transmitting medium. A ruby chip was included in the DAC so the pressure could be determined from the shift of the R1 emission line [46]. The incident X-ray radiation from the synchrotron was of wavelength 0.4113 Å on beamline ID15B and 0.3738 Å on ID27. Calibrations on the two beamlines were performed using silicon and CeO_2_, respectively.

X-ray diffraction patterns were collected following stepwise increases of the pressure, with the pressure recorded before and after each step to obtain an average pressure during the measurement. Compression was performed until evidence of pressure-induced amorphisation was observed, upon which the samples were depressurised and found to return to their ambient structure. At each pressure point three 2D diffraction images were taken, which were then averaged and normalised using the FIT2D program [47]. This 2D image was subsequently integrated using the Dioptas software [48], to produce a 1D diffraction pattern.

The unit cell parameters were determined from the diffraction patterns by Pawley refinement in the *P*6_3_/*mmc* space group in the TOPAS Academic software [49]. Sequential refinements were performed for the pressure measurements using the batch-analysis mode. The bulk modulus for each sample was calculated using the PASCal web application [37] by fitting the unit cell parameters measured within the 0–2.2 GPa range to the 2nd-order Birch–Murnaghan equation.

### 3.3. Computational Modelling

Density-functional theory (DFT) calculations were carried out on the empty FAU and EMT frameworks using the plane-wave pseudopotential code VASP [50]. It is impractical to model the disordered Na_x_(Al_x_Si_1−x_)O_4_ structures, particularly given the poorly-defined location of the Na^+^ cations, therefore we constructed models of the 144-atom primitive cell of FAU and the 288-atom cell of EMT based on SiO_4_ tetrahedra as an approximation. As noted in the results, calculations performed on these models yielded similar structural and mechanical properties to the experimental measurements on the empty structures, giving us some confidence in this approximation.

Quantum-mechanical exchange and correlation were modelled using the PBEsol generalised-gradient approximation functional [51] with the semi-empirical DFT-D3 dispersion correction [52] (i.e., PBEsol + D3). The electronic structure was represented using a plane–wave basis with a cutoff of 800 eV at the Gamma point in reciprocal space. Ion cores were modelled by projector augmented-wave (PAW) pesudopotentials [53,54], the O 2s/2p and Si 3s/3p electrons being included in the valence shell. The structures were fully optimised to reduce the magnitude of the forces on the ions to <10^−2^ eV Å^−1^, and a tolerance of 10^−8^ eV was applied to the total energy during the electronic minimisation. The precision of the charge–density and augmentation grids was set automatically to avoid aliasing errors, and the PAW projection was performed in real space.

To simulate the effect of the framework cages being occupied by solvent molecules, we also performed calculations using the implicit-solvent VASPsol model [40,41] to replace the vacuum in the voids with a dielectric continuum with the dielectric constant of water (ε = 80.1).

To obtain the bulk moduli, a series of calculations were performed in which the optimised structure was compressed or expanded in 1% volume increments and then re-optimised at fixed volume before fitting the energy/volume curves to the appropriate form of the Birch–Murnaghan equation of state [55]. As a comparison, we also evaluated the full elastic-constant matrix of FAU using the finite differences method implemented in VASP with a step size of 10^−2^ Å [56].

Lattice dynamics calculations were carried out on the optimised unit cell of EMT and the optimised primitive and equivalent conventional cells of FAU using the Phonopy package [57] with a finite-displacement step size of 10^−2^ Å. Phonon density of states (DoS) curves and thermodynamic functions were calculated based on phonon frequencies interpolated onto a regular Γ-centered grid of q-points with 24 × 24 × 24 subdivisions. Simulated infrared (IR) spectra were obtained by combining the Γ-point phonon frequencies and eigenvectors with Born effective-charge tensors calculated with the density-functional perturbation theory (DFPT) routines in VASP [58], using the open-source SpectroscoPy package [59,60].

For the calculations of the elastic constants, phonon frequencies and Born charges, the PAW projection was applied in reciprocal space to ensure accurate forces.

### 3.4. Data Availability

All data created during this study is available free of charge from the University of Bath data archive [61].

## 4. Conclusions

The high-pressure measurements performed in this study confirm our previous predictions that the presence of 18C6 occluded in the EMT topology does not influence the flexibility of the framework. Although compression of the cell volume under pressure is unchanged, the occupation of the *t-wof* cavity by the 18C6 guest helps to maintain structural integrity and resist the onset of pressure-induced amorphisation at higher pressures, which occurs in the empty framework from approximately 4–8 GPa. This result confirms present understanding that 18C6 does not behave as a geometric template for zeolite EMC-2 synthesis, but rather is likely to subtly influence the free energy of crystallisation, encouraging the assembly of the EMT framework over the competing FAU structure.

First-principles calculations were carried out to evaluate the energy differences between the FAU and EMT frameworks and were found to reproduce the measured mechanical properties to a high degree of accuracy. While a comparison of the lattice energies indicates FAU to be more stable than EMT, calculations using an implicit-solvent model suggest that the stability ordering could be heavily influenced, and even perhaps reversed, by the presence of solvent and/or by guest molecules in the cages. Lattice dynamics modelling further suggests that the FAU structure has a more favourable lattice energy, but EMT has a higher vibrational entropy, resulting in it being stabilised by free energy at finite temperature. The latter of these results raises the interesting possibility that the role of 18C6 may be to suppress the formation of a kinetic product and allow a more stable phase to nucleate and/or grow, posing an important question to be addressed in future experimental work.

Our results also reinforce the utility of geometric simulations in predicting framework flexibility when species are occluded within the characteristic cavities, and emphasise the partnership between geometric simulations, high-pressure structural data and computational modelling for understanding the behaviour of framework materials.

Overall, the combined findings from these experimental and theoretical studies highlight the complexities of structure direction during zeolite crystallisation, in particular the way in which major topological changes can be propagated through subtle variations in the synthesis scheme. When selecting organic additives for the rational design of new zeolites, our results show that geometric matching is not the only factor that needs to be taken into account. We propose that the use of space-filling organic additives is not necessarily the correct approach to prepare new zeolites, but rather that organic additives should be designed to push the crystallisation process to explore the metastable parts of the free-energy landscape.

## Figures and Tables

**Figure 1 molecules-24-00641-f001:**
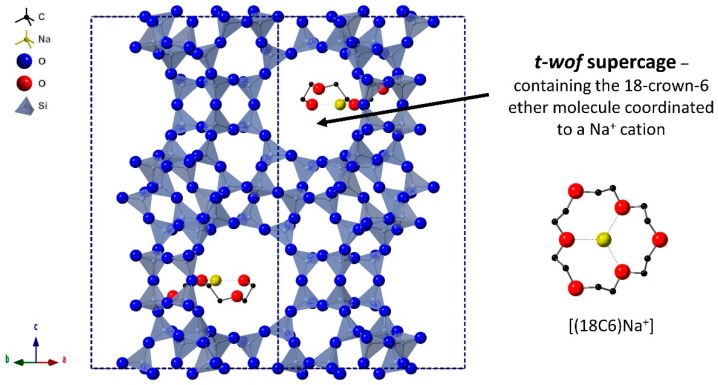
Crystal structure of the zeolite EMC-2 (EMT) framework along the {110} direction. The positions of the 18C6 molecule in the *t-wof* supercages, in the form of the [(18C6)Na^+^] macrocation, as determined by Baerlocher et al. [28] are highlighted.

**Figure 2 molecules-24-00641-f002:**
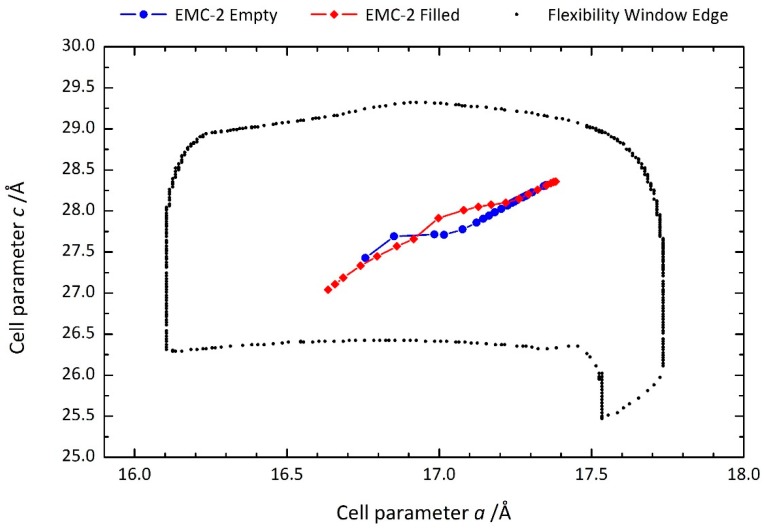
Unit cell dimensions of filled (red diamonds) and empty (blue circles) crystalline zeolite EMC-2 (EMT) under applied pressure. The predicted boundaries of the flexibility window of the EMT framework from previous geometric simulations [25] is overlaid for comparison.

**Figure 3 molecules-24-00641-f003:**
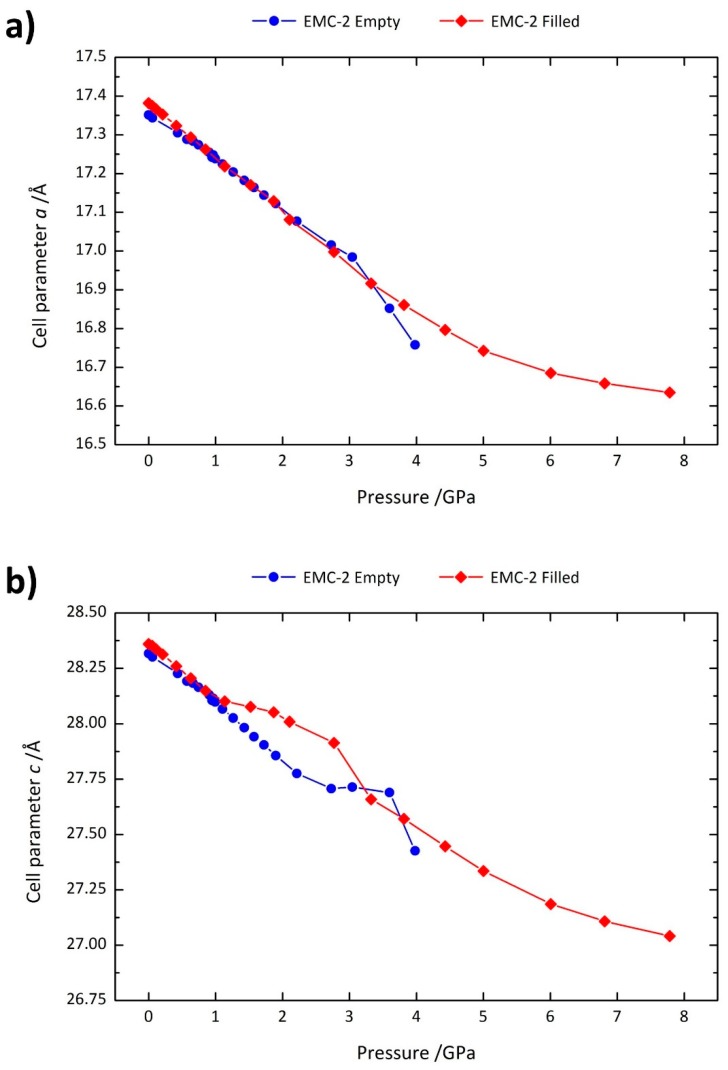
Variation of the (**a**) *a* and (**b**) *c* axis lengths of zeolite EMC-2 (EMT) as a function of pressure. Data for filled and empty structures are shown as red diamonds and blue circles, respectively.

**Figure 4 molecules-24-00641-f004:**
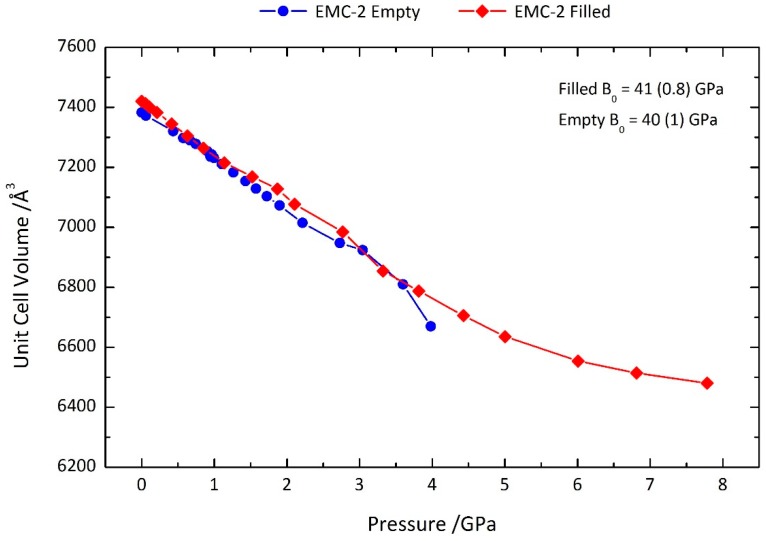
Unit cell volume of zeolite EMC-2 (EMT) as a function of pressure. Data for the filled and empty structures are shown as red diamonds and blue circles, respectively. Also shown are the bulk moduli (B_0_) of the two zeolites obtained by fitting the volume/pressure curves to the 2nd-order Birch–Murnaghan equation using the PASCal web application [37].

**Figure 5 molecules-24-00641-f005:**
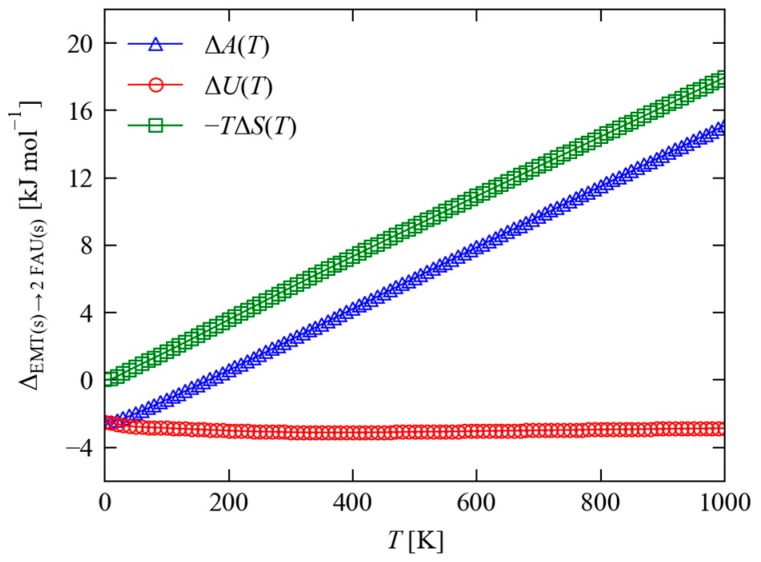
Calculated constant-volume (Helmholtz) free energy ΔA=ΔU−TΔS (blue) for the transformation in Equation (1) as a function of temperature. The red and green curves show the breakdown ΔA into the ΔU and −TΔS terms, respectively. Note that the ΔU term captures both the differences in lattice energy $\Delta U_{\mathrm{latt}}$ and the difference in the vibrational internal energy $\Delta U_{\mathrm{vib}}$ due to the thermal population of the phonon modes, the latter of which is weakly temperature dependent.

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
