# Peer review of "Intrinsic Flexibility of the EMT Zeolite Framework under Pressure"

_molecules, 2019, doi:10.3390/molecules24030641_

Round 1
Reviewer 1 Report
The manuscript “Intrinsic flexibility of the EMT zeolite framework under pressure” by Nearchou et al” reports a combined experimental and computational study of EMT-type zeolites. In the experimental part, in-situ diffraction experiments are used to study the compressional behaviour of EMT samples containing organic templates (18C6 molecules) and of guest-free samples. The computational part focusses on a DFT study of EMT and its comparison to the structurally related FAU framework. The results are sound and should be of interest to the zeolite community, and the manuscript is well written. My only general criticism is that I find that the link between the experimental part and the DFT part is a bit “weak”, and it could be clarified in some instances how the complementary use of both approaches improves the overall understanding.
Apart from this, I have a few minor remarks that the authors should consider when revising the manuscript (in order of appearance in the manuscript):
(0) Abstract
a) Page 1, line 21: The abbreviation “18C6” may be obvious, but it should still be introduced properly.
b) Page 1, line 23: What is “first-principles comparative periodic DFT”? Does this relate to the comparison between EMT and FAU? If so, I think the word comparative should be used in another place.
(1) Introduction
a) Page 1, line 33 and 36: Given that there are many other materials with zeolite topologies that are referred to as “zeolites”, “zeotypes”, or “zeolite-like materials”, the definition of zeolites as aluminosilicates might be too narrow.
b) Figure 1: In the original Baerlocher et al. structure, the 18C6 molecule is disordered. The authors seem to have removed the disorder – thereby lowering the symmetry – without comment in the preparation of the figure. The disorder should either be shown, or they should add a comment that it is omitted for some reason. Moreover, I would suggest to display the O atoms of the 18C6 molecule in a different colour than the framework oxygen atoms.
c) Page 3, line 83: It is not very clearly defined what is meant by “true geometric templating”, maybe this could be clarified, possibly citing an example where it is established that this “true geometric templating” occurs.
(2) Results and discussion
Figures 2, 3, and 4: All these figures suffer from rather poor quality. In part, this may be due to the PDF conversion process, but there are other issues that can easily be resolved: First, I would suggest using a different colour scheme that is better distinguishable. Second, I would encourage the authors to enhance the symbol size, or to use filled symbols, to improve the visibility.
Page 5, paragraph starting line 131: The discussion here is a bit hard to follow. In particular, it is not clear why different P-transmitting media were used, i.e. why was daphne oil not used for the filled zeolite? Moreover, the authors should note that it has been shown by Angel et al. that the hydrostatic behaviour of silicone oil breaks down at pressures as low as 1 GPa – which might indeed explain their observations (DOI: 10.1107/S0021889806045523).
Page 6, line 185: Is the energy difference given per SiO2 formula unit or per unit cell? I suspect the latter, although the former would be more customary in thermochemistry studies. If I am right, it should be clarified to which unit cell it refers (48 or 96 T atoms?), and the value per SiO2 formula unit should be included in brackets.
Page 7, line 216: The free-energy difference is temperature dependent, so it should be stated to which T the value of 2.39 kJ mol-1 corresponds.
Figure 5: Why is DeltaU temperature-dependent? (slight decrease at low temperatures)
Page 7, line 242: Looking at the computed spectra given in Figure S8, I find it rather difficult to envisage a distinction on this basis. If the authors want to maintain their claim, they should mention the distinguishable features in the spectra. In this regard, it could be helpful to visualize the spectrum a second time, with the intensity scale only up to 0.2 (i.e. cutting off the highest peak).
References:
Ref.s 15, 22, and 59 are incomplete, the former two to an extent that it would be difficult to find the article. Furthermore, there are some typesetting issues.
Supporting information:
The PBEsol-D3 unit cell parameter a of FAU reported in Table S1 must be wrong, I suspect a copy-paste error from the previous line (the quoted value is the same as in the previous line despite the different volumes).
Author Response
Dear Editors,
Thank you for sending us the referee reports and we are grateful to the referees for sending their reports quickly. It was pleasing to see that both referees liked the manuscript and suggested some constructive changes. We have revised the manuscript accordingly and have made following changes:
Referee 1:
The manuscript “Intrinsic flexibility of the EMT zeolite framework under pressure” by Nearchou et al” reports a combined experimental and computational study of EMT-type zeolites. In the experimental part, in-situ diffraction experiments are used to study the compressional behaviour of EMT samples containing organic templates (18C6 molecules) and of guest-free samples. The computational part focusses on a DFT study of EMT and its comparison to the structurally related FAU framework. The results are sound and should be of interest to the zeolite community, and the manuscript is well written. My only general criticism is that I find that the link between the experimental part and the DFT part is a bit “weak”, and it could be clarified in some instances how the complementary use of both approaches improves the overall understanding.
To address the general criticism, we have modified the revised manuscript to try and more strongly link the experimental and modelling results. In particular, we have clarified at the start of the section that the purpose of the modelling is to establish the energy differences between the structures, and we have reinforced the main conclusion from the studies at the end of the section. We have also made some small revisions to the conclusions to ensure the experiment and theory are linked together in this section. We hope these modifications will strengthen the paper and make the overall narrative easier to follow.
Apart from this, I have a few minor remarks that the authors should consider when revising the manuscript (in order of appearance in the manuscript):
(0) Abstract
a) Page 1, line 21: The abbreviation “18C6” may be obvious, but it should still be introduced properly.
This was an oversight on our part. We have corrected this, so that at first mention of “18C6” in the abstract the abbreviation is explained. It now reads as: “18-crown-6 ether (18C6)” – as it does when it is first mentioned in the main text.
b) Page 1, line 23: What is “first-principles comparative periodic DFT”? Does this relate to the comparison between EMT and FAU? If so, I think the word comparative should be used in another place.
We thank the reviewer for highlighting the potential for confusion with this terminology. In the revised manuscript, we have changed the sentence to “This was further studied by computational modelling using solid-state density-functional theory and lattice-dynamics calculations.”, which we hope is clearer.
(1) Introduction
a) Page 1, line 33 and 36: Given that there are many other materials with zeolite topologies that are referred to as “zeolites”, “zeotypes”, or “zeolite-like materials”, the definition of zeolites as aluminosilicates might be too narrow.
We thank the reviewer for highlighting this. The beginning of the introduction has been edited so it reads as follows:
“Zeolites are crystalline inorganic polymers, characterised by their open framework structure and microporosity. Typically, zeolites are aluminosilicates however the term is broadly used to describe materials of varying chemistry with zeolitic topologies.”
b) Figure 1: In the original Baerlocher et al. structure, the 18C6 molecule is disordered. The authors seem to have removed the disorder – thereby lowering the symmetry – without comment in the preparation of the figure. The disorder should either be shown, or they should add a comment that it is omitted for some reason. Moreover, I would suggest to display the O atoms of the 18C6 molecule in a different colour than the framework oxygen atoms.
We have now added a few sentences explaining that in the true crystal structure the 18C6 species is disordered and explained why we have lowered the symmetry in the figure – which we hope makes this distinction clearer. The following sentences have been added:
“In the Baerlocher crystal structure the 18C6 species express disorder, whereby there is a superposition of two molecules that are associated by a mirror plane and orientated in opposing directions along the c axis. However, based on the occupancy of the crystallographic sites there is only a single 18C6 molecule per t-wof cavity. To illustrate this, Figure 1 shows the two t-wof cavities in the unit cell occupied by a single 18C6 species which is in one of the two available orientations. This reduced symmetry structure has been used previously as the input for geometric simulations [25].”
We have also edited the figure, so that the oxygen atoms of the 18C6 molecule and framework are different colours as suggested. This is now red and blue respectively.
c) Page 3, line 83: It is not very clearly defined what is meant by “true geometric templating”, maybe this could be clarified, possibly citing an example where it is established that this “true geometric templating” occurs.
We thank the reviewer for pointing out this confusion. We have removed the adjective “true” in the revised manuscript, as we worry that this may be part of the source of confusion. However, the definition of “geometric templating” is contained in lines 53-58 in the revised manuscript. The concept of a “symmetric” as well as molecular recognition between a template and the zeolite framework has been added, which we hope aids in clarifying this term. We have also included a previously cited reference in this explanation, where examples of geometric templating are discussed.
(2) Results and discussion
Figures 2, 3, and 4: All these figures suffer from rather poor quality. In part, this may be due to the PDF conversion process, but there are other issues that can easily be resolved: First, I would suggest using a different colour scheme that is better distinguishable. Second, I would encourage the authors to enhance the symbol size, or to use filled symbols, to improve the visibility.
We thank the reviewer for noting this. In the revised manuscript we have increased the symbol size, line width and changed the colour scheme used in figures 2, 3 and 4, as suggested. We hope that this has improved the visibility of the data in these figures.
Page 5, paragraph starting line 131: The discussion here is a bit hard to follow. In particular, it is not clear why different P-transmitting media were used, i.e. why was daphne oil not used for the filled zeolite? Moreover, the authors should note that it has been shown by Angel et al. that the hydrostatic behaviour of silicone oil breaks down at pressures as low as 1 GPa – which might indeed explain their observations (DOI: 10.1107/S0021889806045523).
We thank the reviewer for pointing this out. In the revised manuscript, we have included the suggested reference and clarified that it is the breakdown of the hydrostatic behaviour of the silicone oil that is leading to the anisotropic compression – as this was the point we were attempting to make. This is further evidenced from the datasets in the SI.
Regarding why different pressure-transmitting media have been used, these were the media available when the measurements were made on the respective beamlines at the ESRF facility. Prior to the data collection it was thought that the different pressure-transmitting media would not significantly influence the unit cell contraction – as both Daphne and silicone oil are non-penetrating, which was our primary concern. However, even after seeing the anisotropic cell parameter contraction for the measurement in silicone oil, it has no substantial influence on the overall cell volume which was our most important observation.
Page 6, line 185: Is the energy difference given per SiO2 formula unit or per unit cell? I suspect the latter, although the former would be more customary in thermochemistry studies. If I am right, it should be clarified to which unit cell it refers (48 or 96 T atoms?), and the value per SiO2 formula unit should be included in brackets.
We thank the reviewer for pointing out this omission. The value quoted is per EMT unit cell, i.e. the energy corresponding to the reaction in Eq. 1. We have clarified this in the text, and given the value per SiO2 formula unit as suggested by the reviewer. We have also done the same for the other values quoted later in the text.
Page 7, line 216: The free-energy difference is temperature dependent, so it should be stated to which T the value of 2.39 kJ mol-1 corresponds.
The free-energy difference was calculated at 300 K, which has been stated in the revised manuscript.
Figure 5: Why is DeltaU temperature-dependent? (slight decrease at low temperatures)
is the difference in lattice energy and the vibrational internal energy due to occupation of the phonon modes (i.e. ). While is temperature independent in the constant-volume model we are using, is weakly temperature dependent, resulting in the small temperature dependence shown in Fig. 5. We have clarified this in the figure caption in the revised manuscript.
Page 7, line 242: Looking at the computed spectra given in Figure S8, I find it rather difficult to envisage a distinction on this basis. If the authors want to maintain their claim, they should mention the distinguishable features in the spectra. In this regard, it could be helpful to visualize the spectrum a second time, with the intensity scale only up to 0.2 (i.e. cutting off the highest peak).
We thank the reviewer for their helpful suggestion. The features we were referring to are the collection of bands from 300-500 cm-1, which we have clarified in the text along with the caveat “whether this is ultimately possible will depend on a number of factors, chief among which is the presence of spectral signals from the solvent and/or 18C6 molecules”. As suggested, we have also modified Figure S8 to include an inset with an expansion of the 300-500 cm-1 region to highlight the differences in band positions and intensities.
References:
Ref.s 15, 22, and 59 are incomplete, the former two to an extent that it would be difficult to find the article. Furthermore, there are some typesetting issues.
We thank the reviewer for pointing out this oversight. We have corrected these three references, in addition to the typesetting issues found in other references.
Supporting information:
The PBEsol-D3 unit cell parameter a of FAU reported in Table S1 must be wrong, I suspect a copy-paste error from the previous line (the quoted value is the same as in the previous line despite the different volumes).
We thank the reviewer for pointing this out. The value has been corrected in the revised manuscript, and we also checked the other lattice parameters and cell volumes in this table for errors.
Reviewer 2 Report
The manuscript reports on a top-qulity research of pronounced importance regarding the structure and flexibility of zeolite structures. The results were precisely interpreted and spectacularly supported by appropriate experimental and complementary theoretical methods including carefuly selected high-level DFT calculations.
Author Response
Dear Editors,
Thank you for sending us the referee reports and we are grateful to the referees for sending their reports quickly. It was pleasing to see that both referees liked the manuscript and suggested some constructive changes. We have revised the manuscript accordingly and have made following changes:
Referee 2:
The manuscript reports on a top-qulity research of pronounced importance regarding the structure and flexibility of zeolite structures. The results were precisely interpreted and spectacularly supported by appropriate experimental and complementary theoretical methods including carefuly selected high-level DFT calculations.
We thank the referee for their comments.